# Three-Dimensional Printing Using Biomass–Fungi Composite Materials: Brief Retrospective and Prospective Views

**DOI:** 10.3390/bioengineering11080840

**Published:** 2024-08-17

**Authors:** Zhijian Pei, Al Mazedur Rahman, Brian D. Shaw, Caleb Oliver Bedsole

**Affiliations:** 1Department of Industrial & Systems Engineering, Texas A&M University, College Station, TX 77843, USA; almazedurrahman@tamu.edu; 2Department of Plant Pathology and Microbiology, Texas A&M University, College Station, TX 77845, USA; brian.shaw@ag.tamu.edu (B.D.S.); olib@tamu.edu (C.O.B.)

Petroleum-derived plastic materials are used to manufacture a wide range of products. Approximately 8.3 billion metric tons of plastic products have been manufactured worldwide, which is roughly 1.6 metric tons for every square kilometer of the earth’s surface [1,2]. The majority of these products end up as waste, with approximately 300 million metric tons generated annually and 4.8 to 12.7 million metric tons entering the ocean [3]. Planning a sustainable future requires selecting alternatives to petroleum-derived plastic materials. Biomass–fungi composite materials offer a viable alternative. The biomass materials used in these biomass–fungi composite materials are usually agricultural waste (such as wheat straws, corn stover, and wood dust) and biodegradable. However, they are often undervalued and misused. In these biomass–fungi composite materials, fungi grow through the biomass particles and bind them together [4,5]. The fungi species used in the relevant studies reported in the literature include *Pleurotus ostreatus*, *Ganoderma lucidum*, and *Fomes fomentarius*. The main advantages of biomass–fungi composite materials include their biodegradability and less negative impact on the environment in comparison to petroleum-derived plastic materials [6] and their good thermal and sound insulation [7,8]. They can be used to manufacture many products (that are traditionally made from petroleum-derived plastic materials) in the packaging, construction, and furniture industries [9]. At the end of their life, they can be recycled into new products. Additionally, they can be produced locally and used for space habitation.

Early investigations demonstrated that biomass–fungi composite materials exhibited adequate mechanical strength and biodegradability [10]. Subsequent studies were conducted on the optimization of their growth and processing conditions to enhance the physical properties of biomass–fungi composite materials, as well as the effects of substrate selection on density and strength of biomass–fungi composite materials [11]. Comparative studies of different fungal species, such as *Ganoderma lucidum* and *Pleurotus ostreatus*, showed that the growth dynamics of the mycelial network affected the properties of biomass–fungi composite materials [12]. Innovations in manufacturing processes included the use of molds and heat treatment to enhance the structural integrity of biomass–fungi composite materials [13]. Currently, molding-based methods [14] and hot-pressing methods [15] are used to manufacture products using biomass–fungi composite materials. However, these methods consume a lot of energy, offer limited flexibility during product design, and are usually not cost-effective for producing small quantities of products.

Three-dimensional printing-based manufacturing methods offer several advantages over traditional molding-based and hot-pressing methods in manufacturing products using biomass–fungi composite materials. Firstly, 3D printing allows for the creation of products with complex geometries and customized shapes. Secondly, 3D printing eliminates the need for a costly and time-consuming step—making molds for each design. Thirdly, 3D printing can reduce material waste by depositing material only where it is needed. Fourthly, 3D printing can potentially lower energy consumption by eliminating the need for the high-temperature and high-pressure processing steps typical of hot pressing. Additionally, 3D printing can streamline the production process by integrating multiple steps into a single operation, thereby reducing the overall production time and associated costs.

Although 3D printing-based manufacturing methods are commonly associated with small batch production and prototyping, they provide significant benefits for specific scenarios within industries such as packaging, construction, and furniture. In the packaging industry, 3D printing-based manufacturing methods can be used for creating customized packaging solutions and the rapid prototyping of new designs. In the construction industry, 3D printing-based manufacturing methods allow for the fabrication of complex architectural elements and customized components that would be difficult or costly to produce using traditional methods. In the furniture industry, 3D printing-based manufacturing methods enable the creation of customized structures catering to individual customer preferences.

Preliminary experimental results [16] showed that fungi could survive the mixing and printing stages (of 3D printing-based manufacturing methods) and continue growing on the printed samples. More studies were reported on the effects of mixture composition on the rheological properties of biomass–fungi mixtures prepared for 3D printing [17]; the effects of waiting time (between the time the mixture was prepared and the time of 3D printing using the mixture) on the mechanical and rheological properties of biomass–fungi mixtures; the minimum printing pressure required for continuous extrusion during 3D printing and the quality of printed samples (characterized by layer height shrinkage and filament width uniformity) [18]; and the effects of crosslinking (using sodium alginate and calcium chloride) on fungal growth and viability for the 3D printing of biomass–fungi composite materials [19].

There have been other reported studies on 3D printing-based manufacturing methods using biomass–fungi composite materials. Bamboo fibers and chitosan were included in biomass–fungi composite materials to improve the printability and mechanical properties of 3D printed parts [20]. Shredded cardboard and natural gums were used to maintain the structural integrity of printed parts [21]. Mycelium fibers and alginate microsheets were used to enhance the mechanical properties of 3D printed parts through freeze-drying [22]. A dedicated extrusion system for robotic printing was developed, and investigations were conducted on the rheological and biological behavior of biomass–fungi composite materials [23]. Another study focused on 3D printing workflows and developing extrudable mixtures with mycelium-based composites cultivated on wastepaper and waste cardboard, along with exploring methods to mitigate contamination [24]. A versatile hydrogel formulation for the intricate 3D printing of *Pleurotus ostreatus* fungi was designed using food-grade reagents such as malt extract, carboxymethylcellulose, cornstarch, and agar. This workflow did not require a sterile environment and could produce structures with additives such as sawdust [25].

Three-dimensional printing using biomass–fungi composite materials faces unique challenges in comparison to 3D printing using plastic, metal, or ceramic materials. For example, if the temperature of a manufacturing process is too high, the fungi might not be able to survive and will not be able to bind the biomass particles together; if the manufacturing environments are not well controlled, contamination can occur, and fungi will not grow properly; and the printed samples usually shrink after drying, affecting geometrical accuracy. Biomass–fungi composite materials often have lower mechanical strength and durability than plastic, metal, or ceramic materials, limiting their applications. Scaling up their production can be difficult due to the biological nature of the materials and the need for controlled growth conditions. Overcoming consumer perceptions and gaining market acceptance for products made from biomass–fungi composite materials is challenging [26,27,28].

To address these challenges, several strategies can be implemented. Standardizing the composition and processing conditions of biomass–fungi composite materials can enhance property consistency. Advanced techniques such as the genetic engineering and selective breeding of fungi can improve growth consistency and reduce contamination risks. Enhancing the mechanical properties of biomass–fungi composite materials can be achieved by incorporating reinforcing agents or optimizing composite compositions. To overcome consumer perceptions and gain market acceptance, extensive education and marketing efforts are necessary to highlight the environmental benefits and unique properties of biomass–fungi composite materials. Demonstrating successful applications and obtaining certifications can also foster trust and lead to acceptance from consumers and industry.

To make 3D printing-based manufacturing methods viable in industry, a wide range of research is needed. The input variables include those related to pre-printing processes, printing processes, and post-printing processes. Pre-printing processes involve substrate preparation, fungal strain selection, and mixture formulation. Post-printing processes include drying and surface finishing. The response variables include printability, cell viability, physical properties, recyclability, and biodegradability. Printability assesses the material’s ability to print complex geometries and maintain the shape of printed structures. Cell viability evaluates the ability of fungi to survive and grow during and after manufacturing processes (such as printing). Physical properties such as mechanical strength, thermal insulation, and sound absorption determine the material’s suitability for various applications. Recyclability examines the potential of reusing or recycling products made from biomass–fungi composite materials at the end of their service lives. Biodegradability measures the environmental impact of the products they break down.

Below are some future research topics.

Numerous fungi species have been explored for molding-based manufacturing methods using biomass–fungi composite materials. However, only *Pleurotus ostreatus*, *Ganoderma lucidum*, and *Fomes fomentarius* have been used in reported studies on the 3D printing of biomass–fungi composite materials. The potential to incorporate thousands of other fungi species (one example would be *Trametes versicolor*) into these biomass–fungi composite materials remains largely untapped. Additional species could potentially offer faster fungal growth and/or improved mechanical properties, enhancing the performance of 3D printed products made from biomass–fungi composite materials for various applications [29].

Various biomass substrate materials have been utilized in molding-based manufacturing methods to make biomass–fungi composite materials. However, the substrate materials investigated for the 3D printing of biomass–fungi composite materials are limited to hemp hurd, waste cardboard, beechwood sawdust, bamboo fibers, chitosan, and shredded cardboard. The exploration of a wide range of biomass substrate materials could significantly enhance the properties of biomass–fungi composite materials. Incorporating diverse biomass materials may lead to improved material characteristics, such as better mechanical strength and improved fungal growth, by providing favorable nutrients for the fungi, thereby broadening the potential applications of 3D-printed products made from biomass–fungi composite materials. Additionally, the density of biomass–fungi composite materials could be tuned based on the substrate materials, which would be advantageous for various applications.

In all reported studies on the 3D printing of biomass–fungi composite materials, predominantly custom-made extrusion-based 3D printers were used. These custom printers have been essential in adapting the extrusion process to accommodate certain unique properties of biomass–fungi mixtures. However, other 3D printing processes could serve as alternatives to extrusion-based 3D printing for biomass–fungi composite materials.

It is worth noting that conducting research on 3D printing-based manufacturing methods using biomass–fungi composite materials often requires expertise from more than one discipline, such as engineering or biology. Future research on the 3D printing of biomass–fungi composite materials should lead to new technology, for example, new compositions of biomass–fungi composite materials, new manufacturing processes for the pre-printing, printing, and post-printing stages; and new designs of products that can be made using biomass–fungi composite materials.

Research on 3D printing-based manufacturing methods using biomass–fungi composite materials (or involving other living matter such as algae and bacteria) presents both challenges and opportunities for manufacturing researchers. This is a great avenue for manufacturing researchers to make an impact on the sustainability of our environment, economy, and society.

## Data Availability

The authors confirm that the analyzed data used to support the findings of this study are available within the article.

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
