# Peer review of "Three-Dimensional Printing Using Biomass–Fungi Composite Materials: Brief Retrospective and Prospective Views"

_bioengineering, 2024, doi:10.3390/bioengineering11080840_

Round 1

Reviewer 1 Report

Comments and Suggestions for Authors

The authors presented a perspective on 3D printing of fungal materials. While the topic is of interest for the bioengineering community, the commentary is very generic with limited scientific arguments about the topic. 

Major criticisms:

1. The first two paragraph has general text for biodegradable materials, which is not inappropriate for a short communication.

2. The authors did not define what is fungal based materials, especially the types (e.g., composites or foams) of fungal materials the manuscript is address.

3. It is scientifically justified why the current manufacturing methods of fungal materials are cost-prohibitive and energy-efficient. 

4. The authors should discuss the limitations of 3D printing for fungal materials and possible ways to overcome them.

5. The authors should provide a vision how 3D-printing can significantly advance the field in the vertical direction.    

Comments on the Quality of English Language

Good

Author Response

Howdy,

The authors appreciate the reviewer’s feedback. They have revised the manuscript according to the comments.

Regards,

Al Mazedur Rahman

Reviewer 2 Report

Comments and Suggestions for Authors

Dear Authors,

The manuscript assumes that biomass-fungi composites offer a promising solution, combining abundant, biodegradable materials with the growth capabilities of fungi to create versatile and environmentally friendly materials. On this basis, they summarize the possibilities of printing using this type of material. After reading the manuscript, I offer some comments for the authors' consideration.

1. The title “3D Printing of Biomass-Fungi Composite Materials: Brief Retrospective and Prospective Views“ is concise but unclear. The authors use the term “material” in the title. In my opinion, the term “product” should be used. Material is a substance mixture that constitutes an engineering object. It's the raw input used in production. The product is the finished item produced from these materials. While the process utilizes materials to create the final object, the result is a tangible, three-dimensional item or product, not a more complex material as an input to other processes. Could “3D Printing of Biomass-Fungi Composite Items (or Products): Brief Retrospective and Prospective Views“ be better?

2. Please consider using “petroleum-derived plastics” instead of “petroleum-based plastics”. While "petroleum-based plastics" is also correct, it focuses more on the raw material itself. The term "derived" highlights the transformation of crude oil into plastic products, essential to understanding the environmental impact and the need for alternative materials.

3. Lines 25-27. The statement “the advantages of biomass-fungi composite materials include low cost, less negative impact on the environment” is only partially true. The “biomass-fungi composite materials” are more expensive than petroleum-derived plastics in the packaging industry. Please revise.

4. The statement "Biomass materials are biodegradable and abundant on this planet" is too broad. While it's true that many biomass materials are biodegradable, the rate of biodegradation can vary significantly. Not all biomass materials are inherently biodegradable. Even products made from biomass can have varying degrees of biodegradability, for example, some highly processed wood products with added chemicals are not biodegradable. Additionally, the availability and abundance of biomass resources differ depending on geographical location. Please rephrase this statement.

5. Disadvantages or challenges of Biomass-Fungi Composite Materials in 3D Printing are missing. While biomass-fungi composites offer promising potential, several challenges and disadvantages are associated with their use in 3D printing: (5.1) Biomass materials can exhibit properties variations, affecting the composite material's consistency and predictability. (5.2) Maintaining consistent fungal growth and preventing contamination during manufacturing can be challenging. (5.3) Compared to traditional materials, biomass-fungi composites often have lower mechanical strength and durability, limiting their applications. (5.4) Scaling up production can be difficult due to the biological nature of the material and the need for controlled growth conditions. (5.5) The current production methods for biomass-fungi composites are less cost-effective than traditional manufacturing processes. (5.6) Overcoming consumer perceptions and gaining market acceptance for products made from unconventional materials is challenging. Please consider mentioning all (or most) these challenges in the text. Specifically, consumer acceptance is the most difficult challenge to overcome.

In essence, the revieved  text serves as a foundation for understanding the potential of biomass-fungi composites in addressing the global plastic crisis. It overviews the challenges and opportunities, inspiring further research and development. Please consider my comments, and good luck with your research plans.

Sincerely

(–)

Author Response

(The authors gave the same response as above.)

Reviewer 3 Report

Comments and Suggestions for Authors

Nice overview of the state of the art and future perspectives for the additive manufacturing of mycelium composites.

Specific comments:

Bottom page 1 to top of page 2: The authors rightfully identify molding-based methods and hot pressing methods as two current drawbacks to mycelium composite production, and then go on to say that 3D printing can overcome these shortcomings. How exactly does 3D printing overcome the shortcomings of post processing methods like hot pressing? Please address.

Additionally, the authors mention that 3D printing is useful for "small batches" or "for producing small quantities of products". This is accurate - 3D printing is mostly used for prototyping at small scale. Is this not also a drawback for the listed applications  on page 1 (which are not small), namely "packaging, construction, and furniture industries?" Please comment on the scenarios where 3D printing makes sense.

Author Response

(The authors gave the same response as above.)

Round 2

Reviewer 2 Report

Comments and Suggestions for Authors

Dear Authors,

thank you for addressing my comments from the first round of review. Among other improvements, thank you for adding a description of the challenges related to mycommaterials. However, after the statement in line 98 ("Overcoming consumer perceptions and gaining market acceptance for products made from biomass-fungi composite materials is challenging") I would expect some reference to literature here. Please consider this comment. Despite this comment, I recommend the manuscript for publication.

Sincerely

Author Response

(The authors gave the same response as above.)
